# On the Role of ROS and Glutathione in the Mode of Action Underlying Nrf2 Activation by the Hydroxyanthraquinone Purpurin

**DOI:** 10.3390/antiox12081544

**Published:** 2023-08-02

**Authors:** Qiuhui Ren, Wouter Bakker, Sebastiaan Wesseling, Hans Bouwmeester, Ivonne M. C. M. Rietjens

**Affiliations:** Division of Toxicology, Wageningen University and Research, Stippeneng 4, 6708 WE Wageningen, The Netherlands; wouter.bakker@wur.nl (W.B.); sebas.wesseling@wur.nl (S.W.); hans.bouwmeester@wur.nl (H.B.); ivonne.rietjens@wur.nl (I.M.C.M.R.)

**Keywords:** purpurin, glutathione, prooxidant, pH-dependence, reactive oxygen species, electrophilicity

## Abstract

Purpurin is a major anthraquinone present in the roots of *Rubia cordifolia* (madder). Purpurin is known to activate Nrf2 (Nuclear transcription factor erythroid 2-related factor 2) EpRE (electrophile responsive element) mediated gene expression as a potential beneficial effect. This study aimed to elucidate the balance between the electrophilicity or pro-oxidant activity of purpurin underlying the Nrf2 induction. For this, Nrf2 activation with modified intracellular glutathione (GSH) levels was measured in an Nrf2 CALUX reporter gene assay. In addition, both cell-free and intracellular ROS formation of purpurin with modified (intracellular) GSH levels at different pH were quantified using the DCF-DA assay. GSH adduct formation was evaluated by UPLC and LC-TOF-MS analysis. GSH and GSSG levels following purpurin incubations were quantified by LC-MS/MS. We show that Nrf2 induction by purpurin was significantly increased in cells with buthionine sulfoximine depleted GSH levels, while Nrf2 induction was decreased upon incubation of the cells with *N*-acetylcysteine being a precursor of GSH. In cell-free incubations, ROS formation increased with increasing pH pointing at a role for the deprotonated form of purpurin. Upon incubations of purpurin with GSH at physiological pH, GSH adduct formation appeared negligible (<1.5% of the added purpurin). The addition of GSH resulted in conversion of GSH to GSSG and significantly reduced the ROS formation. Together these results demonstrate that Nrf2 induction by purpurin originates from intracellular ROS formation and not from its electrophilicity, which becomes especially relevant when intracellular GSH levels can no longer scavenge the ROS. The present study demonstrated that the efficiency of intracellular Nrf2 activation by purpurin and related anthraquinones will depend on (i) their pKa and level of deprotonation at the intracellular pH, (ii) the oxidation potential of their deprotonated form and (iii) the intracellular GSH levels. Thus, the Nrf2 induction by purpurin depends on its pro-oxidant activity and not on its electrophilicity.

## 1. Introduction

Purpurin (1,2,4-trihydroxyanthraquinone) (Figure 1) is a trihydroxy derivative of anthraquinone, first isolated in pure form in 1827 from the roots of *Rubia cordifolia* (Madder plants) [1,2]. Roots of Madder and extracts thereof have been used as a food colorant and red or yellow clothing dye since ancient times [3]. Besides this, the roots of madder are also an important ingredient in traditional Chinese medicine, for preparations used to stop bleeding and stimulate blood circulation to alleviate blood stasis [4]. *R. cordifolia* roots or its extracts are also widely used in India as in traditional medicinal practices for diverse health complaints [5]. Anti-inflammatory, anti-mutagenic, anti-cancer, anti-microbial and neuro-protective potential have been reported for its main constituent purpurin, based on results from both in vitro and in vivo experiments (in rodents) [3,6,7,8]. The mode(s) of action underlying these potential beneficial effects of purpurin remain to be elucidated. Purpurin contains both a quinone and catechol/hydroquinone structural elements; therefore, it can be expected to activate Nrf2 (Nuclear transcription factor erythroid 2-related factor 2)-mediated gene expression. The mode of action underlying this Nrf2 activation by purpurin remains to be investigated and could in theory be linked to either the electrophilicity of its quinone moiety, and/or the pro-oxidant activity of its catechol/hydroquinone moieties, resulting in redox cycling and formation of reactive oxygen species (ROS) (Figure 1).

Nrf2 functions as one of the key players in cellular protective signalling pathways. Under basal conditions, Nrf2 is targeted for ubiquitination and proteasomal degradation by Keap1 (Kelch-like ECH-associated protein 1) in the cytoplasm [9]. In response to electrophilic and/or oxidative stress, Nrf2 dissociates from Keap1 and translocates into the nucleus where it binds to electrophile-responsive elements (EpRE) in the promotor region of target genes, thereby initiating gene expression of the respective protective genes, including, for example, heme oxygenase 1 (HO-1), glutathione S-transferases (GSTs), glutathione peroxidase (GPx) and NAD(P)H: quinone oxidoreductase 1 (NQO1) [10,11,12,13]. Studies on potential modes of action underlying Nrf2 activation have shown that alteration of essential cysteines present in Keap1 by oxidants or electrophiles may be the trigger for release of Nrf2 from Keap1 [14,15,16,17]. This implies that potential modes of action that could explain the Nrf2 activation by purpurin can be related to either its action as prooxidant producing ROS or by acting, via its quinone moiety, as direct electrophile. Either role of purpurin could be affected by intracellular glutathione (GSH) which can scavenge ROS and electrophiles to form oxidized glutathione (GSSG) or adducts, respectively [18]. Meanwhile, GSSG could also modify the sensor cysteines in Keap1 and then activate Nrf2 [19].

The aim of this study was to investigate the mode of action of Nrf2 activation by purpurin. To this end, purpurin-dependent ROS formation, glutathione adduct formation and Nrf2 activation after modification of (intracellular) GSH levels were quantified in cell-free systems at different pH values and in U2OS-based Nrf2 CALUX cells. Given that the negative charge generated in the molecule upon purpurin deprotonation will facilitate ROS formation while hampering thiol reactivity, the reactivity of purpurin in forming ROS and GSH adducts was also studied at a varying pH to facilitate discrimination between the two potential modes of action.

## 2. Materials and Methods

### 2.1. Chemicals and Reagents

Purpurin (purity ≥ 85.0%), Curcumin (purity ≥ 98.0%), L-glutathione reduced (GSH, purity ≥ 98.0%), L-glutathione oxidized (GSSG, purity ≥ 98.0%), L-buthionine-sulfoximine (BSO, purity ≥ 97.0%), *N*-acetyl-L-cysteine (NAC, purity ≥ 99.0%), tert-butyl hydroperoxide (TBHP, 70% in water) 2′,7′-dichlorofluorescein (DCF, purity ~90%) and 2′,7′-dichlorofluorescein diacetate (H2DCF-DA, purity ≥ 97.0%) were obtained from Sigma Aldrich (Zwijndrecht, The Netherlands). All buffer salts, including citric acid, tri-sodium citrate dihydrate, di-sodium hydrogen phosphate dodecahydrate, sodium dihydrogen phosphate monohydrate and formic acid, and acetonitrile for Liquid Chromatography/Mass Spectrometry assay, were also obtained from Sigma Aldrich (Zwijndrecht, The Netherlands). Methanol of analytical grade was ordered from Fisher Scientific (Loughborough, United Kingdom). Dimethyl sulfoxide (DMSO) was purchased from Acros Organics (Geel, Belgium). Dulbecco’s Modified Eagle Medium with Ham’s Nutrient Mixture F-12 (1:1) (DMEM/F12), DMEM/F12 without phenol red, trypsin 0.05% EDTA, nonessential amino acids (NEAA) and phosphate-buffered saline pH 7.4 (PBS), geneticin (G418), penicillin/streptomycin, were purchased from Gibco (Carlsbad, CA, USA). Fetal bovine serum (FBS) was purchased from Invitrogen (Breda, The Netherlands).

### 2.2. Cell Culture

The Nrf2 CALUX cells (BDS, Amsterdam, The Netherlands) that were created based on human osteoblastic osteosarcoma U2OS cells [20] were used in this study. The cells were cultured in DMEM/F12 supplemented with 7.5% (*v*/*v*) FBS, 1% (*v*/*v*) NEAA and penicillin/streptomycin (10 U/mL and 10 μg/mL, respectively) as a growth medium. For selection of cells, 0.2 mg/mL G418 was added once a week. Cells were maintained at 37 °C in a humidified atmosphere with 5% (*v*/*v*) CO_2_. Cells were sub-cultured every three or four days.

### 2.3. Nrf2 CALUX Reporter Gene Assay

The potency of purpurin to activate Nrf2-EpRE mediated gene expression was evaluated by monitoring the stimulation of luciferase activity in the Nrf2 U2OS reporter gene cells. Cells were seeded in the inner 60 wells of 96-well white plates and incubated for 24 h to form confluent monolayers. The seeding density was 3 × 10^4^ cells/well in 100 µL growth medium. Then, the medium was replaced by assay medium (DMEM/F12 without phenol red with 5% (*v*/*v*) DCC-FBS, 1% (*v*/*v*) NEAA and penicillin/streptomycin (10 U/mL and 10 μg/mL, respectively)) containing purpurin (2, 6, 20, 60, 100, 200 µM added to assay medium from a 100-times concentrated stock solution in DMSO) or 1% (*v*/*v*) DMSO as solvent control or curcumin (30 µM added from a 100-times concentrated stock solution in DMSO) as positive control. The plates were kept at 37 °C in a humidified environment with 5% (*v*/*v*) CO_2_. After 24 h exposure, assay medium was removed and 100 µL 0.5 × PBS was added into each well to wash the cells, followed by the addition of 30 µL low salt lysis buffer (containing 10 mM Tris, 2 mM DTT and 2 mM trans-1,2-diaminocyclohexane-*N*,*N*,*N*′,*N*′-tetra-acetic acid mono-hydrate (CDTA), pH 7.8) into each well after removing the 0.5 × PBS. The plates were covered by aluminium foil and kept on ice for 15 min, then stored overnight at −80 °C.

To measure the luciferase activity in the lysates, 100 µL Flash mix (20 mM tricine, 1.07 mM (MgCO_3_)_4_ Mg(OH)_2_, 2.67 mM MgSO_4_, 0.1 mM EDTA, 2 mM DTT, 0.47 mM D-luciferin, 5 mM ATP, pH 7.8) was added into each well. Luciferase activity was measured in relative light units (RLU) using a GloMax-Multi + Micro-plate Mutimode Reader (Promega, Sunnyvale, CA, USA).

The results are expressed as Induction Factor (IF) calculated by dividing the measured RLU of purpurin or curcumin exposed samples by the mean RLU value of the solvent control.

### 2.4. DCF-DA Assay for Intracellular ROS Formation

ROS formation in Nrf2 CALUX cells exposed to purpurin was measured by the DCF-DA assay. The purpurin concentrations tested were selected in line with those used for the concentration response curve for Nrf2-mediated gene expression (2, 6, 20, 60, 100 and 200 µM) and were added from a 100-times concentrated stock solution in DMSO. For the DCF-DA assay, Nrf2 CALUX cells were seeded in the 60 inner wells of black 96-well plates at a density of 3.0 × 10^4^ cells per well in 100 µL growth medium. Then 200 µL PBS was added in each outer well of the plates to limit evaporation from the inner wells. The plates were maintained at 37 °C in a humidified atmosphere with 5% (*v*/*v*) CO_2_ for 24 h to allow cells to form confluent monolayers. Subsequently, the growth medium was removed and cells were washed with 100 µL PBS (37 °C) per well. Following this, supplemented buffer (PBS with 0.4% (*v*/*v*) FBS) containing 25 µM H_2_DCF-DA was added to each well and the cells were subsequently incubated for 60 min at 37 °C in a humidified atmosphere with 5% (*v*/*v*) CO_2_. After removing the H_2_DCF-DA containing buffer, the cells were exposed to purpurin at target concentrations in 100 µL assay medium (DMEM/F12 without phenol red) per well for 4 h. Following this incubation, the fluorescence intensities were measured at λ_exc_ 485 nm and λ_emm_ 535 nm (SpectraMax iD3 (Molecular Devices, San Jose, CA, USA)). Background readings were taken from samples without the addition of H_2_DCF-DA. The results are presented as relative ROS formation (fold increase) in comparison to the solvent control after background subtraction. Cells exposed to 50 µM TBHP were used as the positive control in each plate, inducing a more than 5-fold increase in cellular ROS formation.

To check for the absence of fluorescence quenching in the DCF-DA assay by purpurin, a 2 mM stock solution of the fluorescent dye DCF in ethanol was prepared and stored at −80 °C until use. A solution of 2.5 µM DCF was prepared by dilution from this stock solution in assay medium without FBS. Then 50 µL of this 2.5 µM DCF solution was added to the 60 inner wells of black 96-well plates after which 50 µL assay medium without FBS containing purpurin at two times the target concentration was added [21]. After 4 h incubation at 37 °C in a humidified atmosphere at 5% (*v*/*v*) CO_2,_ fluorescence intensities were measured at λ_exc_ 485 nm and λ_emm_ 535 nm (SpectraMax iD3 (Molecular Devices), San Jose, CA, USA). The results were presented as the fold increase in comparison to the solvent control (1% (*v*/*v*) DMSO).

### 2.5. Modulation of Intracellular GSH Level in the Nrf2 CALUX Cells

Intracellular GSH levels in the Nrf2 CALUX cells were modulated by the addition of either BSO, an inhibitor of γ-glutamylcysteine synthetase leading to a reduction in GSH production, or NAC, a precursor of GSH. Cells were seeded in 96-well white plates as described above. After 24 h growth at 37 °C in a humidified atmosphere at 5% (*v*/*v*) CO_2_, assay medium (DMEM/F12 without phenol red supplemented with 5% (*v*/*v*) DCC-FBS, 1% (*v*/*v*) NEAA and penicillin/streptomycin (10 U/mL and 10 μg/mL, respectively)) containing 30 µM BSO, was added to the cells, which were subsequently incubated for 24 h, in order to decrease their GSH levels. For increasing GSH level, 5 mM of NAC dissolved in assay medium was added for a 4 h pre-treatment of the cells. Subsequently, cells were exposed to assay medium containing purpurin at different concentration in the absence or presence of BSO or NAC for 24 h. Then, cells were washed and lysed, and luciferase activity in Nrf2 CALUX cells was measured essentially as described above.

### 2.6. WST-1 Assay

The cell viability after exposure of the cells to purpurin with or without addition of BSO or NAC was determined using the WST-1 assay. Briefly, cells were plated and treated as described above for the Nrf2 CALUX assay with or without modulation of the intracellular GSH level. Then, 10 µL of WST-1 solution was added into each well. Plates were incubated at 37 °C in a humidified atmosphere with 5% (*v*/*v*) CO_2_ for 1 h, after which the absorbance at 440 nm and 620 nm was measured using a microplate reader (SpectraMax iD3 (Molecular Devices, San Jose, CA, USA)). The amount of formazan dye generated by the cells was estimated by subtracting the measurements at 440 nm from the values at 620 nm. The results are expressed as cell viability (%) in comparison to the solvent control set at 100%. The concentrations of BSO and NAC were selected based on a previous study [22] and were shown not to induce cytotoxic in our study.

### 2.7. Acid Dissociation Constant (pKa) Prediction

The pKa values of purpurin were predicted by MolGpka, obtained through the web server (https://xundrug.cn/molgpka, accessed on 17 August 2022). The SMILES (Simplified Molecular Input Line Entry System) format of purpurin was used as input.

### 2.8. pH-Dependent Incubation of Purpurin with Glutathione

Purpurin (200 µM final concentration added from a 20 mM stock solution in DMSO) was incubated with 200 µM GSH (added from a 20 mM stock solution in nano pure water) in a solution consisting of 40% (*v*/*v*) methanol and 60% (*v*/*v*) 0.1 M citrate or phosphate buffer, the latter depending on the pH which varied from 5.0 to 8.0 [23,24]. Reactions were initiated by the addition of purpurin and upon incubation in a water bath at 37 °C, samples were collected at 0 h, 0.5 h, 1 h, 2 h, 4 h, 6 h, 8 h, 12 h and 24 h. To terminate the reaction, 4 µL acetic acid was added to 196 µL sample [22]. Samples were stored at −80 °C until analysis for GS-purpurin adducts or GSH/GSSG concentration by UPLC or LC-MS/MS.

### 2.9. Ultra-Performance Liquid Chromatography (UPLC) Analysis of Purpurin Glutathione Adduct Formation

The adduct formed upon incubations of purpurin with glutathione was analyzed using a UPLC Nexera series (Shimadzu, Kyoto, Japan) equipped with a Photodiode Array (PDA) detector. The adduct and purpurin were separated on a Phenomenex C18 column (50 × 2.1 mm, 1.7 µm) and detected at a wavelength ranging from 200 nm to 400 nm. The injection volume was 3.5 µL. Mobile phase A was water containing 0.01% (*v*/*v*) trifluoroacetic acid (TFA). Mobile phase B was 100% acetonitrile. The total run time was 15 min, starting with 10% B, and increasing to 100% B in 9 min, maintaining 100% B for 1 min before returning to 10% B over 30 s, and maintaining 10% B till end of the run. The column temperature was kept at 40 °C and the auto-sampler at 10 °C. Purpurin and its glutathione adduct were detected and quantified at wavelength λ 257 nm, using a calibration curve of purpurin.

### 2.10. Liquid Chromatography-Time of Flight-Mass Spectrometry (LC-TOF-MS) Analysis of Purpurin Glutathione Adduct Formation

An Agilent 1200 series high performance liquid chromatography system coupled with a Bruker microTOF (time-of-flight mass analyzer) (Bruker Daltonics, HB, Germany) was used to qualitatively detect the adduct of purpurin with GSH. A phenomenex C18 column (50 × 2.1 mm, 1.7 µm) fitted with C18 precolumn (Phenomenex, Torrance, CA, USA)was used. Water with 0.1% formic acid (A) and 100% acetonitrile with 0.1% formic acid (B) were used as the mobile phase at a flow rate of 0.3 mL/min. The elution gradient was same as used in UPLC analysis: 0–9 min, 10–100% B; 9–10 min, 100% B; 10–10.5 min, 100–10% B; 10.5–15 min, 10% B. The injection volume was 10 µL. Mass spectrometric analysis was performed in negative electrospray ionization mode, with an *m*/*z* range from 100–1500.

### 2.11. Liquid Chromatography Mass Spectrometry (LC-MS/MS) Analysis of GSH and GSSG

LC-MS/MS analysis was performed to assess concentrations of GSH and GSSG [22]. An injection volume of 0.2 µL for analysis of GSH and 1 µL for the analysis of GSSG was used for the incubation samples as well as for the calibration samples. Samples were loaded on a Phenomenex Luna Omega Polar C18 column (100 mm × 2.1 mm, 1.6 µm). Mobile phase A was water containing 0.1% formic acid, mobile phase B was 100% acetonitrile containing 0.1% formic acid. The run consisted of a linear gradient of 0–65% B over 5 min, followed by isocratic elution at 65% B for 2 min, a second linear gradient of 65–80% B over 0.1 min, followed by isocratic elution at 80% B for 9 min, and a final wash with 100% A for 9 min. The mobile phase eluting from the column was introduced into the mass spectrometer between 2 and 11 min for GSH and from 3.5 to 11 min for GSSG measurements. The temperature of the column was 25 °C and that of the autosampler 10 °C during the LC-MS/MS run.

Mass spectrometric analysis was performed using a triple quadrupole mass spectrometer (Shimadzu LCMS-8045, Shimadzu, Japan), fitted with an electrospray ionization (ESI) source operated in multiple reaction monitoring (MRM) in positive mode. The MRM for GSH were *m*/*z* 307.9 > 179.05 (CE −12 eV); 307.9 9 > 76.1 (CE −25 eV); 307.9 > 162.05 (CE −16 eV); 307.9 > 84.05 (CE −21 eV) and those for GSSG were *m*/*z* 613.15 > 484.15 (CE −22 eV); 613.15 > 177.10 (CE −22 eV); 613.15 > 355.05 (CE −22 eV); 613.15 > 231.00 (CE −30 eV). GSH and GSSG in the samples were quantified using calibration curves created using commercially available reference compounds.

### 2.12. DCF-DA Assay for Cell-Free ROS Formation of Purpurin in the Absence or Presence of GSH at Different Concentrations and Different pH Values

To assess the intrinsic activity of purpurin to generate ROS, and the influence of pH on this capacity a cell-free DCF-DA assay was used.

In this assay, H_2_DCF-DA first had to be deacetylated to H_2_DCF that is able to interact with ROS resulting in fluorescence. To this end, 0.5 mL 5 mM H_2_DCF-DA was added to a flask containing 2.5 mL methanol and 10 mL 0.01 M NaOH. After 30 min of stirring at room temperature in the dark (flask covered in aluminum foil), 37.5 mL 33 mM NaH_2_PO_4_ was added to terminate the reaction. The final solution contained 50 µM H_2_DCF. The solution remained stable for 2 weeks at 4 °C in the dark [21].

To quantify ROS formation by purpurin, 50 µL of the 50 µM H_2_DCF solution was added to the 60 inner wells of black 96-well plates, after which 50 µL of a solution containing purpurin at two times the target concentration (400 µM, final concentration 200 µM) with or without GSH (final concentrations 5 mM and 10 mM) was added. The solvents used were the same as those used for the pH-dependent incubations with GSH, consisting of 40% (*v*/*v*) methanol and 60% (*v*/*v*) 0.1 M citrate or phosphate buffer, depending on the pH, which varied from 5.0 to 8.0. The final volume per inner well was 100 µL. All outer wells were filled with 200 µL PBS. After 4 h incubation at 37 °C in a humidified atmosphere with 5% (*v*/*v*) CO_2,_ fluorescence intensities were measured at λ_exc_ 485 nm and λ_emm_ 535 nm (SpectraMax iD3 (Molecular Devices), San Jose, CA, USA). Background readings were taken from samples incubated without H_2_DCF. The results are presented as fold increase in comparison to the solvent control at the respective pH value.

To evaluate the efficacy of GSH to scavenge ROS formation induced by purpurin, the DCF-DA assay under cell-free conditions was used. Fifty µL of the 50 µM H_2_DCF solution was added to the 60 inner wells of black 96-well plates, after which 50 µL of solvent consisting of 40% (*v*/*v*) methanol and 60% (*v*/*v*) 0.1 M phosphate buffer at pH 7.4 containing purpurin at two times the target concentration (400 µM, final concentration 200 µM) with or without GSH (final concentrations 0.1 mM, 0.5 mM, 1 mM, 5 mM and 10 mM) were added. The final volume per inner well was 100 µL. All outer wells were filled with 200 µL PBS. After 4 h incubation at 37 °C in a humidified atmosphere with 5% (*v*/*v*) CO_2,_ fluorescence intensities were measured at λ_exc_ 485 nm and λ_emm_ 535 nm (SpectraMax iD3 (Molecular Devices), San Jose, CA, USA). Background readings were taken from samples incubated without H_2_DCF. The results are presented as fold increase in comparison to the solvent control at the respective pH value.

### 2.13. Data Analysis

LC-MS/MS data acquisition and analysis were performed with LabSolutions software 5.98 (Shimadzu, Japan). ChemDraw 18.0 (PerkinElmer, Waltham, MA, USA) was used to draw chemical structures. Concentration response curves and column graphs were produced by Graphpad Prism 5 (San Diego, CA, USA). For statistical analysis a one-way ANOVA analysis with post-hoc Tukey test was performed using Graphpad Prism 5 (San Diego, CA, USA). Statistical significance was defined as *p* < 0.05.

The deprotonation percentage of purpurin at different pH values was calculated by using Equations (1) and (2).
(1)$$pH=pKa+\log\frac{\left[A^{-}\right]}{\left[HA\right]}$$
where [A^−^] represents the deprotonated form and [HA] represents the protonated form and pKa is obtained by MolGpka according to the SMILE of purpurin. The % deprotonated purpurin present at a given pH was subsequently calculated using Equation (2).
(2)$$Deprotonation\;percentage=\frac{\left[A^{-}\right]}{\left[A^{-}\right]+[HA]}$$

## 3. Results

### 3.1. Nrf2 Activation and Intracellular ROS Formation of Purpurin

Figure 2 shows the concentration-dependent induction of Nrf2-mediated gene expression as well as of ROS formation in the Nrf2 CALUX cells exposed to purpurin. Curcumin at 30 µM was used as a positive control since it is a well-established Nrf2 activator, reported before to result in substantial Nrf2 induction in the Nrf2 CALUX assay (IF > 25) [20,25]. A clear concentration-dependent purpurin-mediated induction of Nrf2-mediated gene expression was observed. With increasing concentration of purpurin, the Nrf2 induction increased and reached significance at concentrations at and above 100 µM, with IF values that amounted to around 15 and 45 at 100 and 200 µM of purpurin, respectively. However, none of the purpurin concentrations tested (from 2 µM to 200 µM) induced an elevation of intracellular ROS levels above the levels induced by the solvent control, while ROS formation was about 5.5-fold increased upon exposure of the cells to the positive control TBHP. To ascertain that the lack of detectable ROS formation was not due to quenching of the DCF fluorescence by purpurin. DCF was incubated with increasing concentrations of purpurin. The results thus obtained (Appendix A) showed that purpurin at concentrations ranging from 2 µM to 100 µM did not affect the fluorescence intensity, while at 200 µM purpurin, the fluorescence intensity was reduced to 0.86 compared to solvent control, pointing at a limited effect on fluorescence intensity. Thus, the absence of ROS formation in the purpurin-exposed Nrf2 CALUX cells cannot be due to fluorescence-quenching by purpurin.

### 3.2. Nrf2 Activation and Cytotoxicity of Purpurin in Nrf2 CALUX Cells with and without Modulated Intracellular GSH Levels

To better understand the mechanism of Nrf2 activation by purpurin, the Nrf2 CALUX assay was performed with cells with modulated intracellular GSH levels. GSH levels in Nrf2 CALUX cells were modified by (pre)incubation with BSO, an inhibitor of GSH synthesis resulting in a decrease in the intracellular GSH level [26], or by addition of NAC, a precursor of GSH which could increase intracellular GSH levels [26]. Before testing the consequences of modified GSH levels on the induction of Nrf2-mediated gene expression by purpurin, the effect of the BSO and NAC treatment on cell viability was assessed to enable the use of non-cytotoxic concentrations of these modulators (Appendix A). Figure 3A shows the Nrf2 activation of purpurin in Nrf2 CALUX cells without or with modulated intracellular GSH levels by the addition of BSO or NAC. The addition of 30 µM BSO resulted in an increase of the purpurin-mediated induction of Nrf2-mediated gene expression as compared to the samples exposed to purpurin without the BSO treatment. For example, the induction factor of purpurin at 100 µM increased from 30- to 127-fold (323% increase) and reached significance at 200 µM purpurin, were the induction factor increased from 125- to 413-fold (230% increase). However, exposure of the cells to purpurin in the presence of NAC significantly reduced the induction of Nrf2 mediated gene expression by purpurin from 13- to 3.7-fold (72% decrease) at 60 µM; from 30- to 4-fold (87% decrease) at 100 µM; and from 125- to 5.7-fold (95% decrease) at 200 µM. Figure 3B shows the cytotoxicity under the conditions for which the Nrf2-mediated gene expression was presented in Figure 3A. Treatment with purpurin alone showed weak cytotoxicity, reducing cytotoxicity to around 80% at 200 μM, while the treatment of the cells with BSO exacerbated this effect, resulting in 50% cell viability at 200 µM purpurin. The NAC treatment did not significantly affect cell viability.

### 3.3. pKa of Purpurin

To further elucidate the reactivity of purpurin in Nrf2 activation, as either an electrophile or a pro-oxidant inducing ROS production, the pH-dependent effects were tested at pH values around the pKa of purpurin. To this end, first, the pKa values of purpurin were predicted by MolGpKa (https://xundrug.cn/molgpka, accessed on 17 August 2022) (Figure 4). The lowest pKa was obtained for the hydroxyl moiety at C3, amounting to 5.8. A pKa value of 5.8 implies that at a physiological intracellular pH (i.e., pH 7.4), purpurin will be almost completely (97.6%) deprotonated. Thus, it is of interest to study the ROS-forming capacity and electrophilicity of also the deprotonated form of purpurin. Deprotonation of purpurin can be expected to affect its electrophilicity and its potential for redox cycling and ROS formation. This might explain the mechanism behind the induction of the observed Nrf2-mediated gene expression.

### 3.4. pH Dependent Cell-Free Incubation Purpurin and GSH

Figure 5 shows part of the UPLC chromatograms of the incubation of 200 µM purpurin incubated for 24 h with an equimolar concentration of GSH. Formation of a metabolite with a retention time of 4.25 min and a UV spectrum similar to that of purpurin itself is observed. Furthermore, LC-TOF-MS analysis of the mixture revealed the *m*/*z* of this metabolite to amount to 560.1, reflecting the formation of a GS–purpurin adduct. Quantification of this peak in the UPLC chromatogram reveals glutathione adduct formation at a level of less than 1.5% of the initial amount of purpurin during the 24 h of incubation at pH 8. The comparison of the full views of the UPLC-UV chromatograms of the incubations of purpurin with and without GSH after 24 h at pH 8 are shown in Appendix A. Similar results were obtained at pH 5, 6 and 7. Thus, it appeared that at all pH values tested, the glutathione adduct formation with purpurin was limited, reflecting a relatively low electrophilicity of purpurin.

Figure 6 presents the levels of GSH and GSSG upon incubations of purpurin with GSH at different pH levels. The results obtained for the solvent control incubations, thus without purpurin, reveal that the formation rate of GSSG from GSH increased with increasing pH, although it occurred at a rate much lower than what was observed in incubations with increasing pH in the presence of purpurin. Figure 6A reveals a limited time-dependent conversion of GSH to GSSG at pH 5, below the pKa of purpurin. One-way ANOVA analysis of the results obtained revealed that at pH 5, there were no significant differences between the solvent control and purpurin treatment in either GSH depletion or GSSG formation at any time point (Figure 6A). However, an increasing rate of GSH to GSSG oxidation was observed with increasing pH values and this effect was more pronounced in the presence than in the absence of purpurin (Figure 6B–D). At pH 6, 7 and 8, incubation of GSH with purpurin showed significant differences from the corresponding solvent control incubated without purpurin in GSH depletion and GSSG formation, with the rate of GSH to GSSG oxidation increasing with the pH. Oxidation of GSH already in the absence of purpurin can be ascribed to its autooxidation at high pH values [27], while the presence of purpurin appears to accelerate this oxidation. The recovery is also presented and reveals full recovery of GSH as GSSG. This result is in line with the observation that in the incubations of purpurin with GSH, a negligible amount of purpurin glutathione adduct is formed. In the incubations of purpurin with GSH, GSH is converted to GSSG at pH-dependent rates, pointing at redox cycling and oxidative stress that appears especially efficient at pH values where purpurin is deprotonated, facilitating its action as a pro-oxidant-forming ROS. The results obtained thus point at ROS formation especially by the deprotonated form of purpurin, which is the dominant form of purpurin at physiological pH.

### 3.5. pH Dependent Cell-Free ROS Formation Induced by Purpurin

To further explain the discrepancy between the cellular incubations where exposure to purpurin did not induce detectable levels of ROS and the in vitro incubations at increasing pH where incubation of purpurin with GSH pointed at substantial redox cycling, ROS formation in these latter incubations was investigated. The results obtained are presented in Figure 7 and reveal that at pH 5, there was no ROS formation in a solution of 200 µM of purpurin, while with increasing pH, purpurin increasingly induced ROS formation with increasing pH, amounting to significantly different 3.1-fold, 5.9-fold and 8.2-fold increases compared to control incubations without purpurin at pH 6, pH 7 and pH 8, respectively. This result suggests that, also at an intracellular pH of 7.4, purpurin can be expected to induce ROS formation. The absence of a detectable ROS formation in purpurin -exposed Nrf2 CALUX cells may then be ascribed to efficient intracellular ROS scavenging, resulting in ROS not only reacting with the thiol moieties of Keap1, activating Nrf2, but also with intracellular GSH, creating GSSG (Figure 6).

### 3.6. Cell-Free ROS Formation in Incubations of Purpurin with GSH at Different pH

The swift conversion of GSH to GSSG in incubations with purpurin at physiological pH (Figure 6) seems in contrast to the lack of ROS formation detected in the Nrf2 CALUX cells exposed to purpurin (Figure 2). So, it was hypothesized that the GSH in Nrf2 CALUX cells might scavenge the ROS, preventing their accumulation and detection. To evaluate this hypothesis, ROS formation was measured in incubations of purpurin in the presence of GSH at different pH values using the DCF-DA assay. Figure 8 shows the pH-dependent ROS formation in cell-free incubations of purpurin in the absence or presence of 5 or 10 mM GSH, concentrations in the range of intracellular GSH levels (0.5–10 mM) [28]. With increasing pH, in the absence of GSH, purpurin showed a pH-dependent increase in ROS formation. In the presence of GSH at both 5 and 10 mM GSH, the significantly elevated ROS production observed at pH 6, 7 and 8 was significantly reduced, corroborating that in the presence of GSH at intracellular relevant levels, ROS accumulation is reduced.

Our results reveal that intracellular levels of GSH could scavenge ROS formed upon exposure of the cells to purpurin. To assess the efficacy of ROS elimination by GSH, ROS formation by purpurin incubated with GSH at different concentrations was measured at physiological pH 7.4. The results in Figure 9 show that already 0.1 mM of GSH could significantly decrease ROS formation induced by purpurin from 12-fold to 7-fold, with this reduction being even more pronounced with increasing GSH levels; until at 5 and 10 mM GSH, ROS formation by purpurin is no longer significantly different from the corresponding control.

## 4. Discussion

Purpurin showed strong induction of Nrf2 activation, and the present study aimed to better understand the mode of action underlying this Nrf2 activation by purpurin. A few earlier studies reported that (hydroxy)quinone moiety-containing molecules could activate the Nrf2 signaling pathway either by acting as an electrophile or as a prooxidant [14,15,16,17,29] (Figure 1). As an electrophile, the quinone moiety could react with the Keap1 sensor cysteine residues, thereby resulting in activation of Nrf2 signaling [29]. This electrophilicity would also be reflected in the formation of glutathione adducts of the quinone type intermediates. The other proposed mechanism relates to the hydroquinone moiety, which could act as a center for redox cycling resulting in ROS formation and, subsequently, induction of Nrf2 via oxidation of the sensor thiols in keap1. Similarly, this reactivity would result in oxidation of GSH to GSSG, which could also modulate the key senor cysteines in Keap1 and induce the formation of disulfide bridges, thereby activating the Nrf2 signaling pathway (Figure 1) [19]. In a previous study, the inherent chemical reactivities of anthraquinones were reported to give rise to a direct reaction with GSH or *N*-acetyl-L-cysteine (NAC, a precursor of GSH) in a buffer solution to generate GSH adducts [30]. For example, after 30 min incubation of emodin and NAC in PBS buffer containing P450 enzyme and NADPH, an adduct of emodin with NAC connected at the C-2 in emodin was shown to be formed [31]. Rhein and aloe emodin were also reported to form adducts with NAC or GSH in PBS buffer in the presence of rat or human liver microsomes [15,32]. All of these three anthraquinones showed the ability to directly bind to thiol groups in buffer containing liver microsomes or cytochromes P450 enzymes. However, the U2OS cell line, which was used in the present study, was reported to not contain detectable levels of cytochromes P450 [20,33]. Other studies reported that anthraquinones showed pro-oxidant properties. For example, aloe-emodin significantly increased ROS production in HepaRG cells after 48 h of exposure from 5 to 40 µM [34]. Rhein induced ROS in a concentration-dependent manner in HepG2 and Huh7 cells upon 24 h of treatment with 100 µM to 200 µM of rhein [35]. Emodin at 50 µM was reported to stimulate ROS generation and GSH depletion in L-02 cells [36]. Purpurin was also reported to induce ROS formation in A549 lung cancer cells at 30 µM upon 24 h of exposure [37].

In the present study, the exposure of Nrf2 CALUX reporter gene cells to purpurin did not result in detectable intracellular ROS formation (Figure 2). This discrepancy may best be ascribed to the fact that different cell types may have different antioxidant statuses [38]. The negative results for ROS formation in the Nrf2 CALUX cells, at concentration > 60 µM where Nrf2 induction was dose-dependent and significantly increased, were shown not to be due to a quenching effect of purpurin on the DCF fluorescence by which ROS is detected (Appendix A). Further studies were performed to elucidate the mode of action underlying this observation. Results of these additional experiments provide support for the hypothesis that intracellular levels of GSH are high enough to efficiently scavenge the ROS formed (Figure 8 and Figure 9), thereby preventing their bioaccumulation to detectable levels. GSH did not prevent ROS formation as such, as reflected by the fact that in incubations of purpurin with GSH, substantial conversion of GSH to GSSG was detected, which was far less in the absence of purpurin (Figure 6). This formation of GSSG in incubations of purpurin with GSH and also the induction of Nrf2 at concentration of 60, 100 and 200 mM without detectable accumulation of ROS, reflects that ROS formation may be kept at physiological low levels without inducing oxidative stress while meanwhile inducing Nrf2-mediated gene activation. This may reflect that scavenging by GSH maintains intracellular ROS concentrations at non-harmful levels, reflecting what is generally referred to as eustress, a condition at which ROS is formed but not associated with adverse effects and oxidative stress, but acting as a physiological redox signaling messenger with important regulatory functions [39], which includes the induction of Nrf2-mediated gene expression. The reactivity of purpurin as a prooxidant giving rise to Nrf2 activation via ROS formation instead of via its electrophilicity was further supported by (i) the fact that in the incubations of purpurin with GSH adduct formation was limited (Figure 5 and Appendix A), and (ii) the fact that ROS formation and purpurin-mediated oxidation of GSH to GSSG was increased by deprotonation of purpurin with increasing pH (Figure 6). This observation is consistent with other studies that report that prooxidant properties of polyphenols such as quercetin, (−)-epigallocatechin gallate (EGCG) were enhanced by increasing pH, because of their deprotonation [40,41,42]. The role of GSH in preventing purpurin-induced Nrf2-mediated gene expression was further evident from the fact that purpurin-mediated Nrf2 induction was significantly increased in Nrf2 CALUX cells in which the intracellular GSH levels were reduced by treatment with BSO, while it was decreased when the cells were exposed to purpurin in the presence of NAC supporting efficient GSH synthesis (Figure 3A). The effect of NAC may not only be related to its role as a precursor for synthesis of intracellular GSH, but may also be in part ascribed to its own capacity to scavenge ROS/electrophiles directly [43].

## 5. Conclusions

Taking together all results and considering that the pKa of purpurin is below the physiological pH, it is concluded that induction of Nrf2-mediated gene expression by purpurin proceeds via its deprotonated form, which displays prooxidant redox cycling activity and produces ROS that leads to increased oxidation of GSH to GSSG, reflecting conditions of oxidative stress that are the trigger for activation of Keap1 and the resulting Nrf2-mediated gene expression. The electrophilicity of the quinone moiety of purpurin appears not to play a prominent role in this Nrf2 activation.

Given this mode of action, it is concluded that the efficiency of intracellular Nrf2 activation by purpurin and other anthraquinones will depend on (i) their pKa and level of deprotonation at the intracellular pH, (ii) the oxidation potential of their deprotonated form and (iii) the intracellular GSH levels. Thus, the Nrf2 induction by purpurin depends on its pro-oxidant activity and not on its electrophilicity.

This novel insight in the mode of action of purpurin may support its use in new therapies, since induction of high levels of ROS in ROS-induced toxic therapies mediated by ROS-elevating agents has been proposed for the treatment of, for example, infections or in anticancer therapy [44,45,46].

## Figures and Tables

**Figure 1 antioxidants-12-01544-f001:**
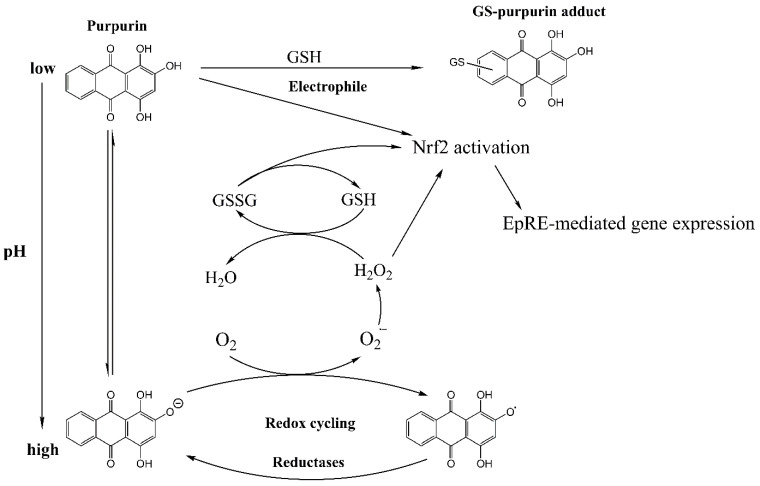
Potential modes of action underlying purpurin-induced induction of Nrf2-mediated gene expression via either electrophilicity or redox cycling and ROS formation.

**Figure 2 antioxidants-12-01544-f002:**
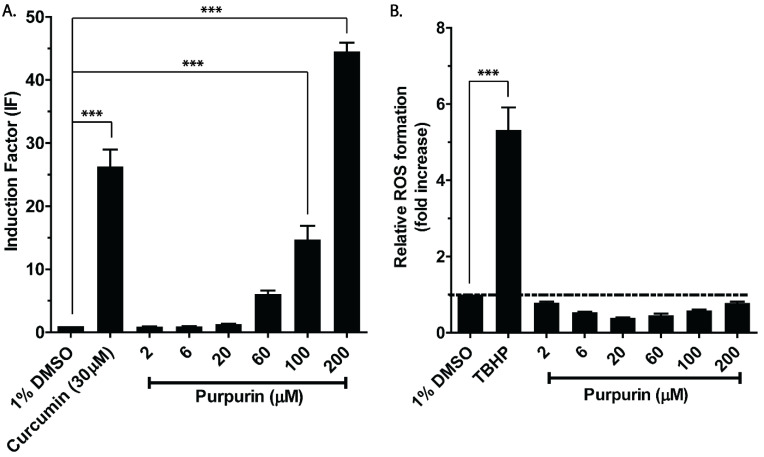
Concentration-dependent effect of purpurin on (**A**) Nrf2-mediated gene expression in Nrf2 CALUX cells treated with purpurin at different concentrations for 24 h and (**B**) intracellular ROS formation in Nrf2 CALUX cells treated with purpurin at different concentrations for 4 h. The ROS formation in the solvent control was set as 1 and indicated by the dashed line. Data are shown as mean fold induction or fold increase compared to solvent control ± SEM of 3 replicates. (***, *p* < 0.001; one-way ANOVA analysis with post-hoc Tukey test).

**Figure 3 antioxidants-12-01544-f003:**
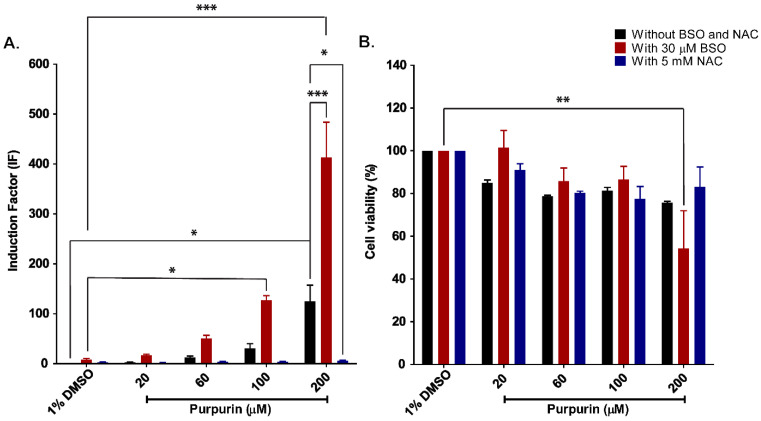
(**A**) Nrf2 activation and (**B**) viability of Nrf2 CALUX cells exposed to 1% DMSO (solvent control) and different concentrations of purpurin in the absence (black bars) or presence of 30 µM of BSO (Red bars), or with 5 mM of NAC (blue bars). The Luciferase induction was expressed as induction factor (IF) compared to the solvent control. The data are presented as mean ± SEM of three independent replicates and Asterisk indicates a response significantly different from treatment of solvent control (1% DMSO) or treatment of purpurin at the respective concentration (*, *p* < 0.05; **, *p* < 0.01; ***, *p* < 0.001; one-way ANOVA analysis with post-hoc Tukey test).

**Figure 4 antioxidants-12-01544-f004:**
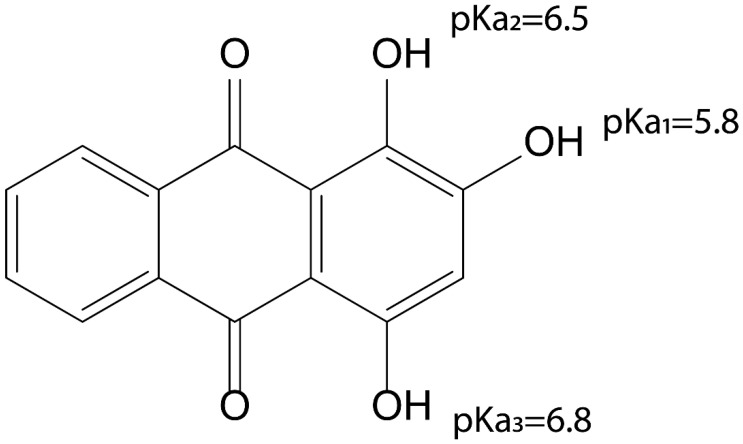
pKa values of purpurin predicted by MolGpka obtained through the web server. (https://xundrug.cn/molgpka, accessed on 17 August 2022).

**Figure 5 antioxidants-12-01544-f005:**
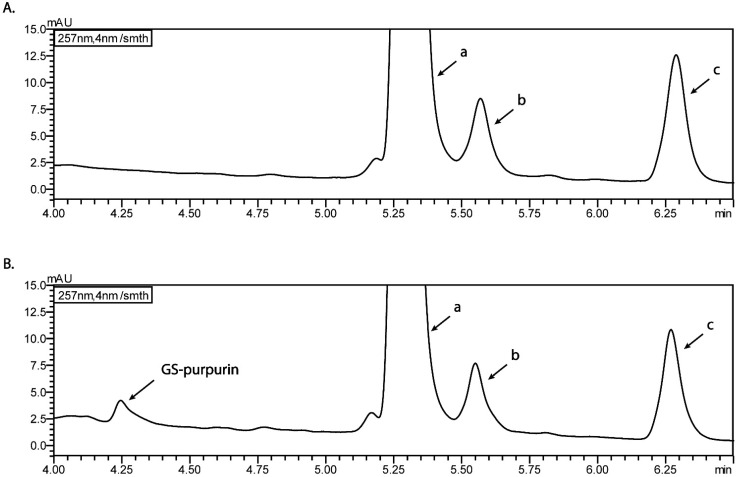
Representative part of the HPLC-UV chromatograms of incubations of purpurin without (**A**) with (**B**) GSH at pH 8. The peaks labeled in a, b, c exist in both incubation samples without or with GSH at pH8 and originate from purpurin, representing purpurin itself (peak labelled a) and impurities present in the commercial purpurin preparation already present at the start of the incubation (peaks b and c). These peaks were not identified to a further extent since the aim of the experiment was to show the (limited) formation of the GS-purpurin adduct. The metabolite eluting at 4.25 min was shown by LC-TOF-MS analysis to have an *m*/*z* of 560.1 representing a glutathione adduct of purpurin.

**Figure 6 antioxidants-12-01544-f006:**
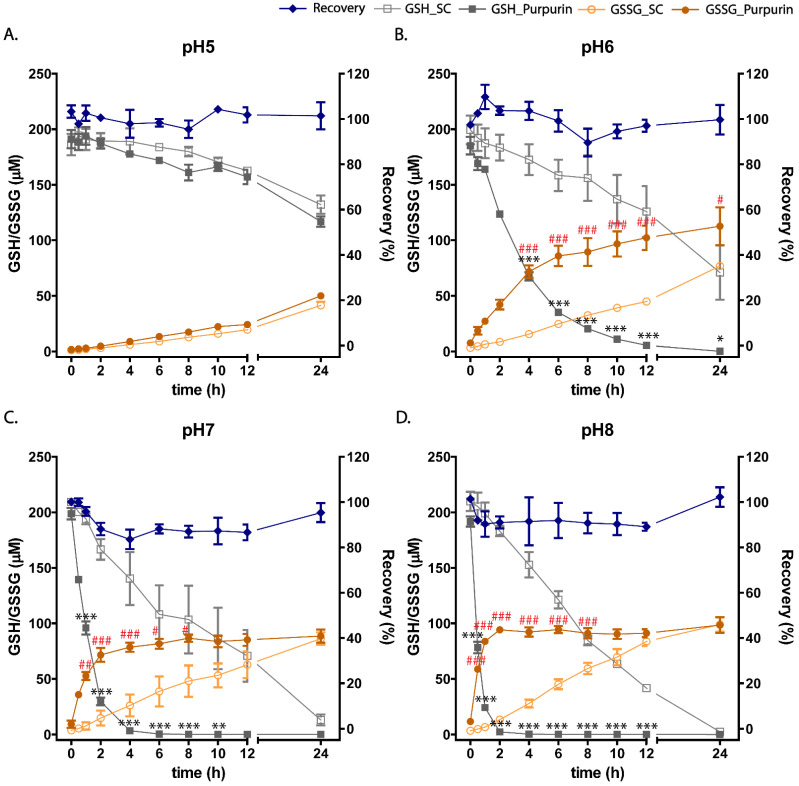
Time-dependent changes in GSH levels, GSSG levels and recovery in incubations of purpurin (200 µM) without or with GSH (200 µM) at pH 5 (**A**), 6 (**B**), 7 (**C**) and 8 (**D**). The data presented as mean ± SEM of at least three independent replicates. GSH_SC: GSH solvent control, GSSG_SC: GSSG solvent control. * in black indicates a response of GSH depletion significantly different from incubations in the absence of purpurin; and # in red indicates a response of GSSG formation significantly different from incubations in the absence of purpurin (* and #, *p* < 0.05; ** and ##, *p* < 0.01; *** and ###, *p* < 0.001; one-way ANOVA analysis with post-hoc Tukey test).

**Figure 7 antioxidants-12-01544-f007:**
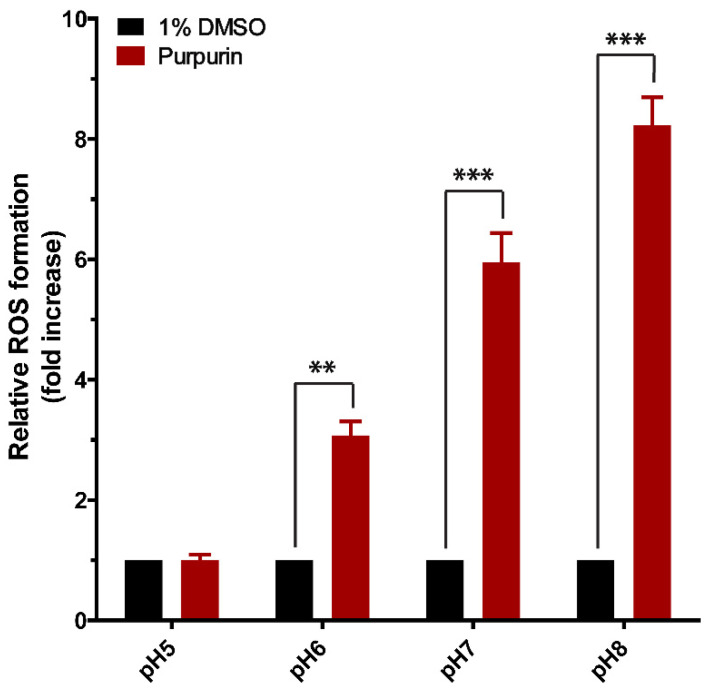
ROS formation in cell-free incubations of purpurin (200 µM) at different pH values. Data are shown as mean fold increase compared to control ± SEM of 3 replicates and Asterisk indicates a response significantly different from treatment of solvent control (1% DMSO) (**, *p* < 0.01; ***, *p* < 0.001; one-way ANOVA analysis with post-hoc Tukey test).

**Figure 8 antioxidants-12-01544-f008:**
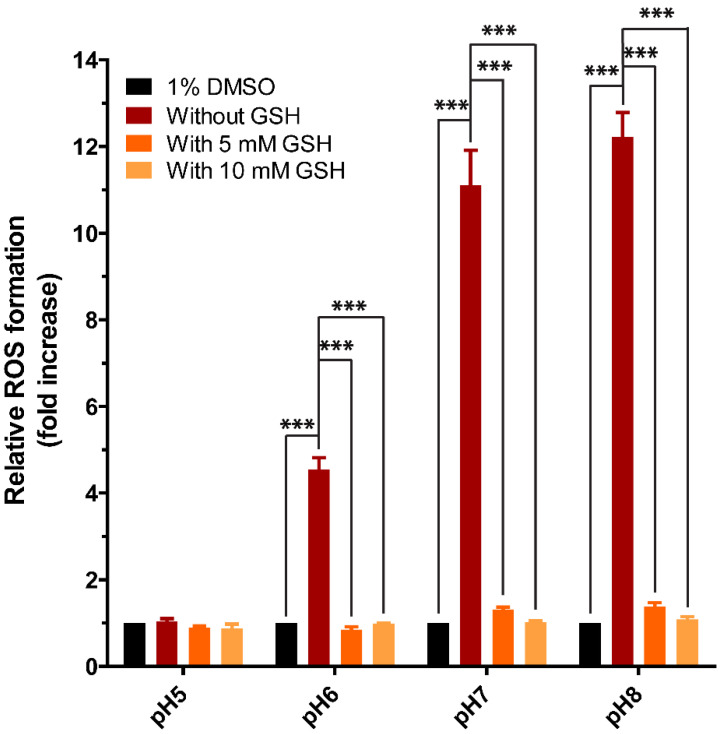
Reactive oxygen species (ROS) measured in a cell-free system with 200 µM purpurin incubated in the absence (Red bars) and presence of 5 mM GSH (Orange bars) and 10 mM GSH (Yellow bars). Data are shown as mean fold increase relative to control ± SEM of 3 replicates and Asterisk indicates a response significantly different from treatment of solvent control (1% DMSO) (black bars) or treatment of 200 µM purpurin alone at different pH (***, *p* < 0.001; one-way ANOVA analysis with post-hoc Tukey test).

**Figure 9 antioxidants-12-01544-f009:**
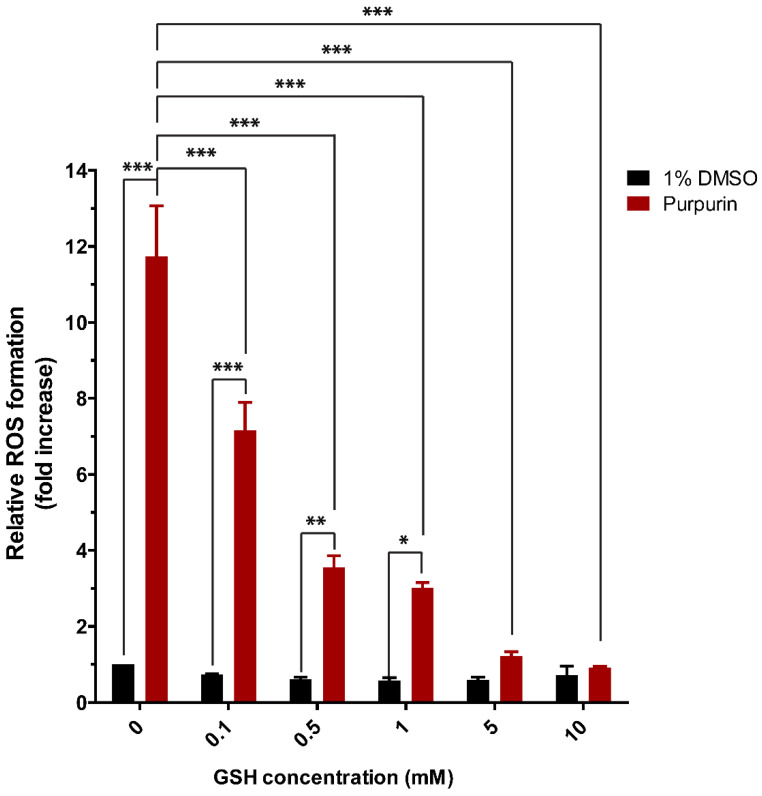
200 µM purpurin induced formation of reactive oxygen species (ROS) in a cell-free system in the presence of GSH at different concentrations (0.1, 0.5, 1, 5 and 10 mM) at pH 5, 6, 7 and 8. Data are shown as mean fold increase relative to control ± SEM of 3 replicates and Asterisk indicates a response significantly different from treatment of solvent control (1% DMSO) or treatment of purpurin alone (*, *p* < 0.05; **, *p* < 0.01;***, *p* < 0.001; one-way ANOVA analysis with post-hoc Tukey test).

## Data Availability

The data presented in this study are available on request from the corresponding author.

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
