# Peer review of "On the Role of ROS and Glutathione in the Mode of Action Underlying Nrf2 Activation by the Hydroxyanthraquinone Purpurin"

_antioxidants, 2023, doi:10.3390/antiox12081544_

Round 1

Reviewer 1 Report

n the introduction to the study problem, the authors clearly presented the underlying problem to be solved. The literature used in the introduction was properly selected. The aim of this study was to investigate the mode of action of Nrf2 activation by purpurine.

Purpurine[a], 1,2,4-trihydroxyanthraquinone is an organic compound, a trihydroxy derivative of anthraquinone. It was first isolated in pure form in 1827 from the root of dyeing marzana. R. cordifolia roots or extracts thereof are also widely used in India, as well as in traditional medical practices for various health ailments. One of the factors influencing the mechanism of action of photosensitizers is their intracellular localization and the resulting target structure. Therefore, the degree and type of induced free radical damage to biologically important molecules is closely related to the structure(s) in which the photosensitive accumulates at the highest concentration (Robertson et. al. , 2009). An important factor that influences the degree of passage of compounds through cell membranes and the intracellular location of photosensitizer is the chemical structure, including, among others: compound charge, hydrophobicity, degree of asymmetry and shape of the molecule (Castano et. al. , 2004). Understanding the mechanisms of action of purpurine may allow the creation of new therapies, especially in the fight against cancer.

Chapter Material and Methods

State-of-the-art techniques were used, including UPLC and LC-MS/MS chromatography and molecular and immunological techniques: Nrf2 CALUX assay, DCF-DA test, WST-1 assay.

In subsection 2. 11. the data of the companies from which GSH and GSSG came should be added.

The statistical methods used are appropriate for the data being analysed.

Chapter Results.

The results were presented in the form of 9 charts, divided thematically into 6 subsections with a description of the results for each. Illustration and description of the presented results are unobjectionable.

Discussion

In the Discussion section, the authors refer to their findings, to the latest research findings in this field. The discussion is conducted efficiently and does not raise any objections from the substantive side. It could only be supplemented by information on the applicability of the results obtained and their use in, for example, biochemistry of medicines or food applications.

Summary of reviews

The authors comprehensively investigated in vitro the effect of purpurine in a variable pH medium on ROS formation in Nrf2 CALUX cells and extracellularly in the presence of glutathione. The data obtained provide valuable insight into the antioxidant or prooxidative activity of purpurine in an in vivo and in vitro environment. The most important information from these studies is that the induction of Nrf2 by purpurine depends on its prooxidative activity and not on its electrophilicity.

After making minor corrections and supplementing the material and methods section, the paper deserves to be published in Antioxidants.

Author Response

Dear  Reviewer #1,

We herewith would like to submit a revised version of our manuscript entitled “On the role of ROS and glutathione in the mode of action underlying Nrf2 activation by the hydroxyanthraquinone purpurin” (ID: antioxidants-2526749) to Antioxidants.

We appreciate the time and efforts that you and the reviewers have taken to critically read and provide valuable feedback on our manuscript, and are grateful for the constructive comments. In attached document we present a point-by-point response to the comments and concerns raised.

Revisions in the manuscript are shown in highlights. The entire manuscript has undergone modification following the journal formatting guideline. We hope the revised version of the manuscript and our accompanying responses will be sufficient to make our manuscript acceptable for publication in Antioxidants.

Looking forward to your response.

With best regards,

Qiuhui Ren

Reviewer 2 Report

1. Reagent supplier information

2. Figure 2, dose-induced differences should be discussed. Why choose Curcumin as positive control and the purity of Curcumin?

3. In Figure 5, although the author has marked the peak of GS-purpurin, other compounds seem to exist in the whole map, and some peaks of the map are also "cut". Therefore, the author is requested to clarify "how the reader should determine the existence of GS-purpurin" and "provide a complete map"

4. Figure 6. Is there any statistical analysis? Is there any difference in the results? Are there any similar literature or statistical results to support the explanation?

5. The design and conclusions of this study are very interesting, but it is suggested that more studies after 2021 should be cited for comprehensive discussion in the discussion.

Author Response

Dear Reviewer #2,

We herewith would like to submit a revised version of our manuscript entitled “On the role of ROS and glutathione in the mode of action underlying Nrf2 activation by the hydroxyanthraquinone purpurin” (ID: antioxidants-2526749) to Antioxidants.

We appreciate the time and efforts that you and the reviewers have taken to critically read and provide valuable feedback on our manuscript, and are grateful for the constructive comments. In attached document we present a point-by-point response to the comments and concerns raised.

Revisions in the manuscript are shown in highlights. The entire manuscript has undergone modification following the journal formatting guideline. We hope the revised version of the manuscript and our accompanying responses will be sufficient to make our manuscript acceptable for publication in Antioxidants.

Looking forward to your response.

With best regards,

Qiuhui Ren

Round 2

Reviewer 2 Report

The author answered my question, I only have one question that needs to be clarified

1. Can add a little more to explain the difference in the influence of purpurin on Nrf2 and ROS when the concentration of purpurin is 60, 100 and 200 in figure 2

Author Response

Dr. Aileen Wang

Section Managing Editor Antioxidants

Dear dr. Aileen Wang,

We would like to submit there-revised version of our manuscript entitled “On the role of ROS and glutathione in the mode of action underlying Nrf2 activation by the hydroxyanthraquinone purpurin” (ID: antioxidants-2526749) to Antioxidants.

We appreciate the time and efforts that you and the reviewers have taken to critically read and provide further suggestion on our manuscript. In attachment, we present our answer to the final comment suggested by reviewer #2.

New revisions in the manuscript are shown in red in order to distinguish highlight marks included in the first revised version of our manuscript. We hope the re-revised version of the manuscript and our accompanying responses will be sufficient to make our manuscript acceptable for publication in Antioxidants.

Looking forward to your response.

With best regards,

Qiuhui Ren
